# Geosynthetics for Filtration and Stabilisation: A Review

**DOI:** 10.3390/polym14245492

**Published:** 2022-12-15

**Authors:** Anna Markiewicz, Eugeniusz Koda, Jacek Kawalec

**Affiliations:** 1Institute of Civil Engineering, Warsaw University of Life Sciences—SGGW, 02-787 Warsaw, Poland; 2Department of Geotechnics & Roads, Faculty of Civil Engineering, Silesian University of Technology, 44-100 Gliwice, Poland

**Keywords:** geosynthetics, polymers, drainage, filtration, stabilisation, civil engineering

## Abstract

Geosynthetics have been commonly used for the construction of civil engineering structures such as retaining wall, road and railways, coastal protection, soft ground improvement work, and landfill systems since the 1960s. In the past 40 years, the development of polymer materials has helped to prolong the life of geosynthetics. In terms of the practical use of geosynthetics, engineers must understand their appropriate application. The first part of this paper provides a basic description of geosynthetics, including their types, components, and functions. The second part deals with the geosynthetics used as filters. This part briefly presents the mechanism of filtration, the factors affecting the durability of geotextile filters, design concepts, laboratory tests, and case studies. The third part of the study covers the use of geosynthetics for stabilisation. Its mechanism was explained separately for geogrids and for geocells. Several examples of applications with geosynthetics intended for the stabilisation function are described in the last part of this paper.

## 1. Introduction

### 1.1. Types of Geosynthetics

A geosynthetic is defined as a product which has at least one component which is made from a synthetic or natural polymer, in the form of a sheet, a strip, or a three-dimensional structure, used in contact with soil and/or other materials as an integral part of a civil engineering structure, project, or system [1]. Over the past 30–45 years, geosynthetics have been commonly used for a wide range of civil engineering applications. This may be due to a number of factors. Geosynthetics are characterised by having a high resistance to chemical and biological degradation, a high flexibility, a long-term durability, and they are non-corrosive. They are also easy to store and transport, are cost effective, and they are environmentally friendly. Importantly, geosynthetics have been added to the list of traditional construction materials such as rock, soil, cement, bitumen, steel, and brick. The global geosynthetics market was valued at USD 11.5 billion in 2022 and is projected to reach USD 37.9 billion by 2030 [2,3,4,5,6,7].

According to ISO 10,318 [1], there are four families of geosynthetics: geotextiles, geotextile-related products, geosynthetic barriers, and geocomposites (Figure 1).

A geotextile is a planar permeable, polymeric textile product in the form of a flexible sheet. Currently, geotextiles (Figure 2) are classified into three categories based on the manufacturing process [1]:A woven geotextile: a geotextile produced by interlacing, usually at right angles, two sets of yarns, filaments, or other elements using a conventional weaving process with a weaving loom;A knitted geotextile: a geotextile produced by interloping one or more yarns, filaments, or other elements together with a knitting machine, instead of a weaving loom;A nonwoven geotextile: a geotextile produced form directionally or randomly orientated fibres, filaments, or other elements by needle-punching, bonding with partial melting, or chemical binding agents.

Needle-punched nonwoven geotextiles are certainly the most important geotextiles to be installed, for example, in drainage systems. Needling is a process of bonding nonwoven web structures by mechanically interlocking the fibres through the web. Barbed needles, mounted on a board, punch fibres into the web and then are withdrawn, leaving the fibres entangled. The needles are spaced in a non-aligned arrangement and are designed to release the fibre as the needle board is withdrawn (Figure 3) [8,9,10,11,12,13,14].

Geotextile-related products (Figure 4) are planar, permeable, and polymeric (natural or synthetic) materials used in contact with the soil and or other materials in civil and geotechnical engineering applications, which do not comply with the definition of a geotextile. There are many geotextile-related products, including [1]:A geogrid: a planar, polymeric product consisting of a regular open network of integrally connected, tensile elements, which may be linked by bonding, extrusion, or interlacing, whose openings are larger than the constituents;A geonet: a planar, polymeric material consisting of a regular dense network of integrally connected parallel sets of ribs overlying similar sets at various angles;A geocell: a three-dimensional, permeable, polymeric (synthetic or natural) honeycomb, or similar cellular structure, made of linked strips of geosynthetics;A geostrip: a polymeric material in the form of a strip of a width no more than 200 mm, used in contact with soil and/or other materials in geotechnical and civil engineering applications;A geomat: a three-dimensional, permeable structure made of polymeric filaments, and/or other elements (synthetics or natural), mechanically, and/or thermally and/or otherwise, bonded;A geospacer: a three-dimensional polymeric structure with an interconnected air space in between it;A geoblanket: a permeable structure of loose, natural, or synthetic fibres and geosynthetic elements bonded together to form a continuous sheet.

Geosynthetic barriers (low-permeability geosynthetic materials, used with the purpose of reducing or preventing the flow of fluid through the construction) are also an established product group in civil engineering and the geoenvironmental industry. They include factory-made polymeric geomembranes, bituminous barriers (bitumen attached to geotextile), and geosynthetic clay liners in the form of a sheet in which the barrier function is essentially fulfilled by clay (Figure 5) [1,2,15,16,17,18].

The combined sheet types of geosynthetics are geocomposite (Figure 6). In geocomposites (drainage, reinforcement, etc.), two or more materials perform specific function more effectively than when they are used separately [1,19,20,21].

### 1.2. Polymers in Geosynthetics

The materials used in the manufacture of geosynthetics are mainly synthetic polymers derived from the rubber, bitumen, crude petroleum oils and fibreglass. Therefore, a variety of polymers can be used to manufacture synthetic fibres and fabrics, including (Table 1) [2,10,22,23,24]:Polyamides (PA) (e.g., nylon-6, nylon-66 and nylon-46);Polyacrylonitrile (PAN);Polyethylene naphthalene (PEN);Polyvinyl chloride (PVC);Polystyrene (PS);Polypropylene (PP);Polyethylene terephthalate (polyester) (PET);Polyethylene (PE, LLDPE, LDPE, HDPE, MDPE).

The most commonly used types are PP, PET, and HDPE [25]. Their structures are shown in Figure 7. Polypropylene was discovered in 1954. It is a semi-crystalline thermoplastic with a density range of 0.9 to 0.91 g/cm^3^. It is a cost-effective raw material and flexible for moulding. However, stabilisers and additives must be added to give the polypropylene UV resistance during processing [26,27]. Polyester is a thermoplastic with a density in the range of 1.22 to 1.38 g/cm^3^. The chemical resistance of PET is excellent. The only exception is very high pH environments. The melting point of PP and PET are equal to 165 °C and 260 °C, respectively [28,29]. Polyethylene is generally used in the manufacture of geomembranes. There are different grades of PE: a linear low density (LLDPE, density 9.20–9.45 g/cm^3^), a low density (LDPE, density 9.2–9.3 g/cm^3^), a high density (HDPE, density 9.40–9.60 g/cm^3^), and a medium density (MDPE, density 9.30–9.40 g/cm^3^). Polyethylene is also a thermoplastic produced from the monomer. The density of polyethylene is higher than polypropylene [24,30,31].

Knowing which polymer is present in the synthetic which has been used is of great importance. There are many factors that affect the durability of polymers. Oxygen, heat, humidity, and an ultraviolet component are the above-ground factors that may cause polymer degradation, while the soil characteristics, temperature, moisture content, heavy metal ions, and alkalinity/acidity belong to the underground factors that can affect a polymer (Table 2). However, polymers’ degradation processes are slow and can be reduced by the application of additives. Ash and carbon black will survive even at temperatures above 500 °C [32,33,34].

### 1.3. Functions of Geosynthetics

Geosynthetics are manufactured for the needs of various civil engineering projects such as a road, dam, the protection of coastal structures, retaining wall, land reclamation, sanitary landfill, foundation, an embankment, as well as a drainage system (Figure 8) [16,35,36,37,38,39,40,41,42,43,44,45,46].

The choice of an appropriate geosynthetic in these applications (specific engineering structures) depends on the functions that the geosynthetic is to perform in it. Geosynthetics perform one or more of the following functions [1,2,7,8,49,50,51]:Reinforcement: the use of the stress–strain behaviour of a geosynthetic material to improve the mechanical properties of the soil or other construction materials;Stabilisation: improvement of the mechanical behaviour of an unbound granular material by including one or more such geosynthetic layers that undergo a deformation under applied loads is reduced by minimising the movements of the unbound granular material;Surface erosion control: the use of geosynthetic materials to prevent or limit soil or other particle movements at the surface of, e.g., a slope;Filtration: the restraining of the uncontrolled passage of soil or other particles subjected to hydrodynamic forces, while allowing the passage of fluids into or across a geosynthetic material;Drainage: the collection and transportation of precipitation, ground water, and/or other fluids in the plane of a geosynthetic material;Separation: preventing the intermixing of adjacent dissimilar soils and/or fill materials;Barrier: the use of a geosynthetic to prevent or limit the migration of fluids;Protection: preventing or limiting the local damage to a given element or material;Stress relief: retarding the development of cracks by absorbing the stresses that arise from damaged pavement (for an asphalt overlay).

The relative importance of each function is governed by the site conditions, and of course the construction application. In many applications, two or more functions of the geosynthetic are required (Table 3).

Apart from the basic functions, geosynthetics can perform some other functions that mostly depend on the basic ones [2];

Insulation: reducing the passage of heat, sound, or electricity;Containment: encapsulating a civil engineering-related material such as rock and soil or a sludge;Absorption: assimilating a fluid;Cushioning: controlling and to damping the dynamic mechanical actions;Surface stabilisation: restrict movement and prevent the dispersion of surface soil particles subjected to the erosion actions of wind or rain;Vegetative reinforcement: extending the erosion control limits.

This paper reviews the developments and applications of geosynthetics in filtration and soil stabilisation. The study also presents the main characteristics, properties, and laboratory tests of selected geosynthetics as well as several case studies using different geosynthetic materials. Additionally, the criteria for the choice of geosynthetics are discussed, which is particularly important for filtration. Therefore, the noticeable impact of the proper selection of geosynthetics on the design service life of civil engineering structures is shown.

## 2. Geosynthetics for Filtration

### 2.1. Mechanism of Filtration

Filtration can be defined as a soil-geosynthetic system in equilibrium that allows for the adequate liquid flow with limited soil loss across the plane of the material over a lifetime of service compatible with the application under consideration [6,10,52,53]. From among all the geosynthetics, nonwoven geotextiles are mainly used as filters. They have many advantages compared to traditional granular filters, such as a low cost, consistent quality, low environmental impact, and convenience of use and installation [54,55,56]. The global nonwoven geotextiles market size is estimated to reach USD 6.7 billion by 2025 [57].

In the filtration process, between the structure of the geotextile and the base soil, a discontinuity arises. It is essential to allow some soil particles to migrate through the geotextile under the influence of seepage flows. A condition of equilibrium should be established immediately after the installation to prevent the soil particles from being piped indefinitely through the nonwoven geotextile. Therefore, three zones (the bridging network, soil filter, and undisturbed soil) may be identified. Geotextiles used for filtration often perform a separation function [2,58].

### 2.2. Factors Affecting on Durability of Geotextile Filters

The compatibility of filtration is predicated on the geotextile satisfying a requirement for soil retention. If it is not ensured, piping or clogging can occur. Piping refers to a soil particle migration through the material and occurs if the pore sizes of the nonwoven geotextile filter are too large. It is not possible to retain the movement of the base soil particles. Clogging is a result of the entrapment of soil particles on or/and within the nonwoven (rarely woven) geotextile (mechanical clogging). This process reduces the permeability of the filter pad.

Palmeira [59] presented three probable causes of physicals clogging: internal clogging, blinding, and blocking (Figure 9). Blocking occurs when coarse particles locate themselves at the entrance of the woven geotextile pores. Blinding happens when the soil particles (mostly fine soil particles) are accumulated near the soil–geotextile interface. The dimensions of the soil particles are smaller than the geotextile pores. Internal clogging refers to the mechanism when the soil particles are entrapment in the geotextile pores. The clogging of the filter can be also due to the biological growth (biological clogging) and precipitation of chemicals: of calcium sulphate, calcium carbonate, or magnesium carbonate, etc. (chemical clogging) [40,60,61,62,63].

The chemical and/or biological clogging of geotextiles, usually used in landfill drainage systems, has been widely studied [60,61,64,65,66,67,68]. Correia et al. [61] pointed out that clogging by ochre may be considered to be a major threat in the behaviour of drainage systems and filters. The main factor affecting the formation of ochre is oxygen, which is available at the water–air interface of the filters. Fleming and Rowe [64] presented the batch test, column test, and field-scale mesocosm test results to investigate the clogging mechanism. They used fresh raw leachate from the Keely Valley Landfill site and the composition of synthetic leachate. It was observed that a reaction between calcium (Ca) and carbonate (CO_3_) usually led to a clog. Other authors also used calcium and carbonate for the preparation of synthetic leachate (Table 4).

Gardoni et al. [66] described the laboratory test results of the water permeability of three needle-punched PET nonwoven geotextiles. The geotextiles were clogged by leachate from the landfill Belo Horizonte. After 30 days of artificial clogging, the water permeability coefficient decreased by approximately 1000 times. Koda et al. [68], by contrast, reported the water permeability test results of nonwoven geotextiles after 12 years of exploitation in the Radiowo landfill drainage system. The water permeability coefficient reduced by 74%.

Apart from the chemical/biological clogging, physical clogging has been analysed by many researchers. These studies can be divided into three categories: the laboratory tests of nonwoven geotextiles artificially clogged [41,73,74], the laboratory tests of nonwoven geotextiles after many years of exploitation in drainage systems [75,76,77], and the laboratory tests of the behaviour of the soil–geotextile system [78,79]. The most common laboratory test used to assess the compatibility of the soil–geotextile is the gradient ratio (GR) [60,80,81]. The gradient ratio is expressed as [82,83]:(1)$$\mathrm{GR}=\frac{i_{\mathrm{sg}}}{i_{s}}$$
where i_sg_ is the hydraulic gradient in the soil–geotextile composite and i_s_ is the hydraulic gradient in the soil.

The gradient ratio value should not exceed 3.0 [83]. It is worth noticing that gradient ratio values increase with time. The results of the GR tests of needle-punched nonwoven geotextiles were presented by Sabiri et al. [54]. One type of soil (MSa) was used in this test. The results obtained show that the value of the GR increased by approximately three times after 140 h of the filtration process (Figure 10).

Zhou et al. [78] carried out gradient ratio tests on the needle-punched nonwoven geotextile and soil sample from the field. The tests were performed under the hydraulic gradient at 4, 7, and 11. It was observed that the gradient ratio values increased not only with time, but also with the hydraulic gradient (Figure 11). Similar test results were presented by Hong and Wu [84].

Most of the research works and gradient ratio tests were performed according to the applicable standard [82], where the soil–geotextile interface is calculated for the segment of the soil specimen between 25 and 75 mm above the nonwoven geotextile filter. However, the determination of the GR should be based on the water head measurements closer to the nonwoven geotextile filter interface, e.g., 4 mm and 8 mm above the geotextile [79]. It is important to predict more accurately the soil–geotextile interaction behaviour. Moreover, air bubbles can be entered into the voids and block the water flow with time. It is essential that the use of deaired water in the experiment reduces the risk of forming air bubbles within the test set-up.

### 2.3. Design Approaches

A geotextile filter must retain soil and let water through. The design approaches to geotextile filters are often group into geometrical and hydraulic criteria. The first one defines the limit values for the void diameters to hinder the transport of smaller particles through them. The second one defines a limit value for the hydraulic gradient at which the transport of the soil particles begins [10].

The proper design and selection of geotextile filters strongly depends on the boundary conditions, on the criticality of the application (e.g., the filters used in embankment dams), and on the geotechnical characteristics of the base soil (e.g., the grain size distribution, internal stability, and permeability). The boundary conditions are closely related to the applied hydraulic gradients, the flow conditions, and the behaviour of the soil–geotextile system. Unfortunately, the commonly used filter design criteria do not consider all these factors but are often the result of a necessary compromise [85,86].

#### 2.3.1. Retention Criteria

According to the retention criteria, the filter should have openings small enough to retain the soil. In the early days, the pore diameters, for which 95% of the remaining pore diameters are smaller and the size of the larger particles of the soil, were compared [87]:O_95_ ≤ d_85_,(2)
where O_95_ is the apparent opening size and d_85_ is the 85% finer grain diameter.

However, retaining only large particles works only if the large soil particles retain the smaller ones. In all other cases, the additional criteria should be considered. The internal stability of the soil is the key parameters. To check the internal stability, the approaches of Kezdi [88], Sherard [89], and Kenney and Lau [90] can be used.

After many laboratory tests of the geotextile filters, the retention criteria were modified by many authors (Table 5) [80,81,91,92]. In general, these criteria can be written using the following formula:O_F_ ≤ R_R_d_n_,(3)
where O_F_ is the geotextile characteristic opening size (e.g., O_98_—the pore diameter for which 98% of the remaining pore diameters are smaller, O_95_—the pore diameter for which 95% of the remaining pore diameters are smaller, O_90_—the pore diameter for which 90% of the remaining pore diameters are smaller, O_50_—the pore diameter for which 50% of the remaining pore diameters are smaller, and O_15_—the pore diameter for which 15% of the remaining pore diameters are smaller); R_R_ is a constant value dependent on the approach criteria; and d_n_ is the indicative diameter of the base soil particles (e.g., d_90_—the grain size diameter of which 90% of the distribution is finer, d_85_—the grain size diameter of which 85% of the distribution is finer, d_50_—the grain size diameter of which 50% of the distribution is finer, d_15_—the grain size diameter of which 15% of the distribution is finer, and d_30_—the grain size diameter of which 30% of the distribution is finer).

Another design proposal was published by Heibaum et al. [104], based on the findings of Giroud [85,97] and Christopher and Holtz [99]. The published criterion follows a shifted lognormal distribution. The formula in the range f(x) = O_90_/d_50_ ≥ 1 and x = C_U_ ≥ 1 is expressed as follows:(4)$$f(x)=1+\frac{18}{0.45\cdot x\sqrt{2\pi }}\exp(-\frac{{\left(\mathrm{lnx}-1.5\right)}^{2}}{2\cdot 0.45^{2}}),$$
and the following criteria are met:O_90_/d_50_ = 1 for C_U_ ≥ 1, according to DVWK [105];O_90_/d_50_ = 5 for C_U_ ≈ 4, according to Giroud [85,97] and CFEM [106];O_90_/d_50_ ≈ 3 for C_U_ = 7 to 8, according to DVWK [105];O_90_/d_50_ ≈ 1 for C_U_ ≥ 20, according to Giroud [85,97], DVWK [105], and CFEM [106].

The comparison between the normalised retention criteria and C_U_, including the described lognormal distribution, is presented in Figure 12. All the criteria agree to use the low ratios of O_90_/d_50_ for a high coefficient of the uniformity values. It is also observed that they accept higher values of the ratio O_90_/d_50_ in the range of C_U_ from 3 to 6.

According to ISO 12,956 [107], all the geotextile filter design criteria in Europe should be based on the characteristic opening size O_90_. This parameter is determined by the manufacturer.

#### 2.3.2. Permeability Criteria

Nonwoven geotextile filters need openings large enough to allow for water to flow almost freely. This could result in the loss of some of the finest soil particles [39,81,85,86].

Calhoun [87] distinguished between the apparent opening size O_95_ and the 15% finer grain diameter (d_15_):O_95_ ≥ d_15_,(5)

Christopher and Holtz published another well-known design proposal (for less critical applications) [99], FHWA [100], Giroud [85], and Cazzuffi et al. [81]:k_n_ ≥ k_s_,(6)
where k_n_ is the coefficients of permeability of the geotextile and k_s_ is the coefficients of the permeability of the soil.

However, many researchers proposed higher values up to k_n_ ≥ 10k_s_ [78,98,100,104] or even up to k_n_ ≥ 100k_s_ [108]. Moreover, Giroud [109] recommended taking into account the hydraulic gradient in the base soil in the vicinity of the geotextile filter (i_s_), e.g., in earth dam cores: k_n_ ≥ 10k_s_i_s_. Based on the experiences, Giroud [85] proposed new permeability criterion:k_n_ ≥ max(i_s_k_s_; k_s_),(7)

The typical values of the i_s_ are presented in Table 6.

#### 2.3.3. Clogging Resistance

Clogging decreases the permittivity and is hard to anticipate. However, during the past four decades, clogging criteria have been reported in the literature. Christopher and Holtz [99] summarised that geotextiles should be not clogged if:O_95_ > 3d_15_ for C_U_ > 3,(8)
O_15_/d_15_ > (0.8 ÷ 1.2),(9)
O_50_/d_50_ > (0.2 ÷ 1.0).(10)

On the other hand, according to Luettich et al. [110], to minimise the risk of clogging, the following criteria should be met:Use the largest opening size (O_95_) that satisfies the retention criteria;For nonwoven geotextiles, use the largest porosity available, but not if it is less than 30%;For woven geotextiles, use the largest percent open area available, but not if it is less than 4%.

The authors recommend that all the above clogging criteria should be closely related with the gradient ratio (GR) and with the number of constrictions (for the needle-punched nonwoven geotextiles). The value of the GR should be less than three, as mentioned in Section 2.2.

The constriction is a window delimited by three or more fibres in which the soil particles could migrate. A soil particle that travels through a geotextile filter moves from one constriction to another, following a filtration path [79,85,109,111]. According to Giroud [85,109] and the ASTM D7178-16e1 standard [112], the number of constrictions is calculated as follows:(11)$$m=\sqrt{1-n}\frac{t_{\mathrm{GTX}}}{d_{f}}$$
where n is the geotextile’s porosity, t_GTX_ is the geotextile’s thickness, and d_f_ is the fibre diameter.

This is a common way to calculate the number of constrictions, however, Elsharief and Lovell [113] and Urashima and Vidal [114] presented another equation (Table 7) for calculating the number of constrictions in nonwoven geotextiles.

It is important to mention that the filtration properties of the nonwoven geotextiles, with the same or similar characteristic opening size, can be different. Delmas et al. [116] studied the soil–geotextile system behaviour of two nonwoven geotextiles (A and B) having the same opening size (O_100_ = 80 µm) but with the number of constrictions equal to 25 and 50, respectively. They reported that the gradient ratio values for the geotextile B were two times greater than for geotextile A. Similar test results were obtained by Miszkowska [48]. To prevent clogging, the number of constrictions should range from 20 to 45 [79].

### 2.4. Applications of Geotextile Filters

The geotextile filter are components of many hydraulic structures, including retaining wall and abutments drainage systems, buried drains as pavement edge drains areas for drains and filters, erosion control systems, slope drainage, landfill leachate collection systems in railway tracks and pavement base course layers, drainage blankets, silt fences, and drains to accelerate the consolidation of the soft foundation soils (Figure 13) [16,36,38,40,43,46,48,49,64,80].

In many applications for railways, roads, dikes, levees, and other embankments, a filter is placed horizontally and then covered by a fill. The loads for this type of installation are traffic and overburden. The overburden of a geotextile filter may influence its performance. Nonwoven geotextiles are compressible, and an increased stress level can significantly have an influence on the loss of permittivity, etc.

On the other hand, drainage trenches are built mostly with vertical walls and a filter lining before filling the trench with permeable material. For that reason, the main requirement in that case is to create a large filter surface. The cross section of the trench should be large. Moreover, in lakes, rivers, canals, and ponds, a bank protection with hard armour is needed.

In using geotextile filters as part of the revetment on slopes, it is vital to ensure that the foundation on which the filter fabric is laid is stable even under the dynamic load conditions imposed by waves [10].

Below, two examples of using geotextile filters in engineering constructions in Poland are presented.

Case study 1

Nonwoven geotextiles can be used in leachate drainage systems. The management of leachates from landfill is one of the most important systems ensuring the protection of the soil–water environment. In Poland, geotextiles were used in the drainage ditches at Radiowo landfill. The landfill is situated in the northwestern part of Warsaw (52°16′37″ N, 20°52′45″ E).

Drainage ditches were made in 2000. The role of the ditches was to collect leachate from the landfill and precipitation water from the slopes. In the landfill, a recirculation system was applied (Figure 14). Leachate was collected through the retention trenches and passed on to storage tanks. Water was pumped to the crown of the landfill.

Drainage ditches have been operated for over 12 years. In 2011, a plan was made to substitute retention diches with a pipe drainage system. Stępień et al. [117] studied the influence of clogging on the filtration behaviour of nonwoven geotextiles after 12 years of exploitation in the Radiowo landfill drainage system. The samples of the analysed nonwoven geotextile were collected during the modernisation works in the landfill. The tests were repeated after 3 years of sample storage in the constant humidity chamber by Koda et al. [68]. The obtained values of the water permeability coefficient for nonwoven geotextile samples (worn and unworn) under different loads are presented in Table 8. It was observed, that after many years of exploitation, the permeability properties of nonwoven geotextiles decreased by almost 4 times. However, the most commonly used permeability criterion (Equation (6)) was still met.

Case study 2

A second example is the earthfill dam Białobrzegi, where nonwoven geotextiles were used in a drainage system. The earthfill dam Białobrzegi is one of eight side dams of the Zalew Zegrzyński Water Reservoir. The Zegrzyński Reservoir was established in 1963.

Initially, the Białobrzegi dam was drained by a drainage pipe (Φ 200 mm, control chamber every 50 m) and discharged into a ditch (every 100 m), but due to the difficult geological conditions of the foundation structures, suffusion occurred; therefore, it was essential to make a renovation. In 1994, PP/PET needle-punched nonwoven geotextiles were used in the drainage system (Figure 15) [48,77].

This is also the case where the laboratory tests of the hydraulic properties of the geotextile filters were performed after many years of exploitation. Miszkowska et al. [118] studied the change in the water permeability coefficients of the clogged geotextile specimens exploited for 23 years in the Białobrzegi earthfill dam drainage system. They reported that the water permeability coefficients decreased by approximately 2.58 times. The obtained tests results are presented in Table 9. Similar test results were obtained by Nieć et al. [119].

The laboratory tests of the nonwoven geotextiles after their exploitation in drainage systems are rare so each of those made by the researchers is valuable. It provides essential information about the behaviour of geotextiles under in situ conditions. The knowledge about the clogging process and the soil–geotextile filter compatibility is important to properly design drainage systems with geotextiles. It is essential to ensure the durability and stability of the construction. Otherwise, drainage failure can occur. Koerner and Koerner [120] presented 69 field failures involving geotextile filters. The authors confirmed that engineers must know the nature of the upstream soil and its permeability. Moreover, a proper installation is also the key to the appropriate functioning of the geotextile filters.

## 3. Geosynthetics for Stabilisation

### 3.1. Mechanism of Stabilisation

Stabilisation is a general term widely used in civil engineering to describe various applications, technologies, or an expected effect. In road engineering, an extremely popular technology for the improvement of weak subsoil is called chemical stabilisation [121]. In geotechnical engineering anchoring, nailing or reinforcing unstable slopes leads to slope stabilisation, understood as the final effect of applying these technics. In the context of geosynthetics, stabilisation is used to define the specific unique function. It is important to remember that the stabilising function was introduced to differentiate from the reinforcing function due to significant differences in the mechanisms and applications which require these specific functions. As mentioned in paragraph 1.3, the stabilisation function of geosynthetics is defined by ISO [1] as an improvement in the mechanical behaviour of an unbound granular material by including one or more geosynthetic layers in such a way that deformation under the applied loads is reduced by minimising the movements of the unbound granular material. However, it is a very generic definition. In practice, an effective stabilising mechanism is achieved due to the lateral resistance provided by the geosynthetic to the aggregate layer which results in its confinement [122]. The natural effect of it is observed as, e.g., an increase in the modulus of such an aggregate layer which improves its load distribution capacity and, consequently, in the reduction in its deformability.

An effective confinement cannot be provided by every geosynthetic. In the literature, stabilisation is often wrongly recognised as a mechanism related not to the confinement but to the tensioning membrane. The tensioning membrane requires large deformations to mobilise the tensile strength of the geosynthetic and this mobilisation increases with the increase in the deformation. Additionally, the material acting as the tensioning membrane requires anchorage to be able to provide support. For these reasons, the tensioning membrane is classified as a mechanism which is related to the reinforcing function (Figure 16).

For pavements, Cook et al. [123], after analysing full scale trafficking trials, introduced a simple classification of the deformations observed on the sub-grade and sub-base surfaces: CON, CON-, 50/50, MEM-, and MEM which could be used to determine the stabilising capability for a specific geosynthetic. In this context, a geosynthetic, to be effective in stabilisation (which, according to Cook et al. [123], is defined as MEM), must be openwork (must have apertures which could be penetrated by grains) and must be stiff (must provide a response against a potential grain movement even at, initially, extremely low deformations expected under an applied load). Among the variety of geosynthetics, there are two groups of products which could provide an efficient stabilisation—geogrids and geocells. The mechanisms of interaction with the aggregate for both are different (Figure 17).

### 3.2. Stabilisation Provided by Geogrid

The stabilising effect of the geogrid is achieved mostly via the interlock of particles in apertures. The surface friction is never dominative due to the much greater proportion of openings in geogrids. The grains placed on the geogrid do penetrate through openings and, when compacted, they increase the shear resistance of the aggregate layer which is normally achievable only via friction. The reduction in the horizontal strain leads to a decrease in the Poisson ratio of the aggregate/geogrid composite material in comparison to the Poisson ratio of the aggregate itself. The reduction in the Poisson ratio increases the horizontal stiffness, which means that the geogrid stabilised soil layer is capable of distributing the vertical stresses on a wider area. The interlocking mechanism increases it mostly within the level where the geogrid is placed up to some distance from it, and the literature calls it the fully confined zone [124]. Then, the efficiency of the confinement becomes reduced up to a distance from the geogrid where the share resistance goes back to the normal value for the aggregate. This intermediate thickness where an increased share resistance goes from a higher value to zero is called the transition zone and the thickness above is called the unconfined zone (Figure 18). The thicknesses of the fully confined and transition zones are in every case depended on the individual properties of the geogrid and aggregates, but also on the compaction technics used and the compaction degree achieved. Grids with a high stiffness in plane together with a hard crushed aggregate whose particles match with the grid opening size, compacted to achieve a high compaction index, are a combination resulting in the high thicknesses of both of the zones. Flexible grids, poor and/or incorrectly selected grain sizes, and an inadequate compaction are factors that reduce the thickness of both zones, resulting in a poor or no stabilisation effect on the unbound aggregates.

### 3.3. Stabilisation Provided by Geocells

The stabilising effect of geocells is achieved mostly due to the limitation of the horizontal deformation of fill material by geocell walls. Under the vertical load cells, circumferential stresses are mobilised which, together with the additional resistance provided by the surrounding cells and the friction of the fill material on the cell walls, reduce the horizontal movements. The system is limited by the strength of the connections of the cells. Due to the on-site installation of the geocells, the efficiency of the stabilisation by geocells is dependent on the bearing capacity of the subgrade; for very weak subgrades, the construction ability is limited. The thickness of the full confined zone is equal to the cell height (Figure 19).

### 3.4. Design Approaches

Design approaches vary by applications and by a specific geosynthetic. However, due to the relatively late recognition of stabilisation as a function and the variability in the geosynthetics intended to use for this function, there is no unanimity for the design methodology. A major part of the known design approaches for geogrids is based on the full-scale testing of a specific product for a specific application, such as in the case of, for instance, Robinson et al. [126], Kang et al. [127], Zheng [128], Jas et al. [129], Esen et al. [130], Marcote et al. [131], and Hornicek et al. [132]. For geocells, there are several modifications of the empirical Giroud–Han method [133], such as Leng and Gabr [134] or Pokharel [135]. The other approaches were summarised and presented by Rimoldi [136].

### 3.5. Applications with Geosynthetics Intended for Stabilisation Function

Stabilisation is a geosynthetic function. However, there are many applications where the overall improvement in the performance of the unbound aggregate layers is observed. Most, but not all, of the applications are related to the trafficked areas. Table 10 below presents the list of the most popular applications where stabilisation as the primary required function of the geosynthetic should be specified to achieve the improvement in the specific aggregate layer.

#### 3.5.1. Example of Mining Subsidence Protection in Roads

Stabilisation requires a geosynthetic in mining areas and pavement since the structures constructed on the surface are additionally challenged [137,138]. Subsidence, being the result of the mining activity, makes the surface unstable. The surface behaves as a wave when the next layer of, e.g., coal is excavated from underground (Figure 20). Any structure is at some point additionally compressed and stretched.

The impact of mining deformation can affect the pavement by reducing the modules and reducing the fatigue life of the pavement. For non-stabilised pavement (i.e., no geogrid), loosening deformations cause a considerable increase in the horizontal deformations due to significant reduction in the layers’ rigidity and the subsequent fatigue life [43]. The solution to protect the pavement against the mining influence is creating geomattresses made from multiaxial stabilising geogrids [137]. The typical cross-section of the mining protective geomattress is described below (the layers are listed from the bottom to the top):Multiaxial geogrid (stabilising function);Crushed aggregate 0/63 mm—thickness 30–40 cm;Multiaxial geogrid (stabilising function);Crushed aggregate 0/63 mm—thickness 30–40 cm.

In the Silesia district of Poland, such mattresses have proven their usefulness when on one of the sections, the measured mining settlement of the whole section was above 1.0 m, but the pavement placed on the top of embankment protected by such a geomattress retained its original geometry (Figure 21) [43].

#### 3.5.2. Example of Stabilised Working Platform

Granular layers are often used in working platforms and beneath the foundations to improve the load spread and bearing capacity on weaker clay soils. The installation of a stiff polymer geogrid within the granular layer can improve the bearing capacity significantly, allowing for thinner granular layers to be installed and bringing savings to the cost associated with the smaller volumes of material. The failure of the bearing capacity involves punching shear through the granular layer and a bearing capacity mechanism in the underlying clay, unless the granular layer exceeds a critical thickness, above which the failure of the shear occurs entirely within the upper layer [139]. Lees [140] developed a design method which allows for the prediction of the required thickness of stabilised aggregate to create a safe platform. Figure 22 shows the elements used to construct the trial platform and Figure 23 presents the verification PLT tests.

#### 3.5.3. Example of Railway Ballast Stabilisation

Ballast stabilisation is an application oriented on the extension of time intervals between the maintenance periods in railways [141,142,143] (Figure 24). The usage of geogrid reduces the speed of the deformation of the layer under applied dynamic loads and postpones tamping with the time.

Hornicek et al. [144] provided the results from their extensive observations of the track, of the sections with and without geosynthetics, carried out on one of the main railway tracks in the Czech Republic. The section with a geocomposite (a multiaxial geogrid laminated to a geotextile) showed a considerable improvement in the modulus measured over the observation period (Figure 25) vs. the sections without an installed geosynthetic. Interesting research was carried out by Penn State University; as reported by Liu et al. [145] an artificial laboratory made stones named SmartRock equipped with sensors which helped to understand the differences in the movements and rotations of a single grain within ballast layer under applied dynamic cyclic loads. SmartRock, in the test, is installed above the geogrid and records the real-time movement of the particles, including their translation and rotation. The results of this research showed a substantial reduction in the particle angular acceleration (Figure 26). It could be concluded that thanks to such a reduction in the movement aggregate layer, a deterioration in the time will also be slowed, which leads to the extension in the serviceability of the ballast over time. Bian [146] also researches the movement of the ballast particles but for high-speed railways.

#### 3.5.4. Example of Unpaved Road Base Stabilisation

Robinson et al. [126] studied the influence of a multi-shape multi-axial geogrid on the full-scale unsurfaced test section of unpaved road (Figure 27). The test section consisted of a 25 cm thick crushed aggregate surface layer placed over a weak clay subgrade. Simulated truck traffic was applied using a load cart outfitted with a single-axle dual-wheel truck gear. The rutting performance and instrumentation response data gathered from the earth pressure cells and single-depth deflectometers were monitored at multiple traffic intervals. It was found that the geogrids improved the rutting performance when compared with an unstabilised section. The calculated traffic benefit ratios ranged from approximately 1.2 at low levels of rutting to approximately 13.0 at higher levels of rutting. The instrumentation response data indicated that the geogrids reduced the measured pressure and deflection near the surface of the subgrade layer. The pressure response data in the aggregate layer suggested that the geogrids redistributed the applied pressure higher in the aggregate layer, effectively changing the measured stress profile with the depth.

#### 3.5.5. Example of Pavement Optimisation

Extending the pavement’s design life or reducing its thickness for a given design life, with the use of geosynthetics in an aggregate base layer, is often referred to as “Pavement optimisation”. Pavement optimisation can be described as obtaining the pavement design objectives at the most economic cost. The design objectives in most cases will be to reach the minimum traffic life requirements, but the objective could also be to meet the construction programme or to meet environmental requirements, such as reducing greenhouse gas emissions during construction. Pavement optimisation has been accepted and used to varying degrees around the world. In the US, the use of geosynthetics for pavement optimisation has been acknowledged and has been in common use for many years, being covered by the AASHTO R50-09 Standard [147]. It states: “Geosynthetics are used in the pavement structure for structural support of traffic loads over the design life of the pavement”. The geosynthetic is expected to provide one or both of these benefits: (1) the improved or extended service life of the pavement, or (2) the reduced thickness of the structural section. To verify these assumptions, a series of trafficking tests were performed to quantify the benefits of using one type of geogrid in flexible pavements according to the requirements of the AASHTO R50-09. The testing of the stabilisation geogrid was conducted at the US Army Engineer Research and Development Centre (ERDC) and consisted of three stages in which multiple full-scale pavement sections were constructed and trafficked [148] (Figure 28). Pavement optimisation with the use of geogrids could bring several environmental benefits [149].

#### 3.5.6. Example of Industrial Floor Stabilisation

Industrial floors are an important part of warehouses and factories. The owners of buildings with industrial floors expect them to have a long service life without the need for maintenance. To construct long-lasting industrial floor, a good subbase becomes critical. From various methods for the improvement of the subgrade and the preparation of the subbase, the stabilising of geosynthetics prove to be one of the most effective methods.

The subsoil may be also Improved by the use of a combination of crushed-stone aggregate and geogrids stabilising this aggregate, which creates a bearing stratum for the flooring. Such an improvement in the flooring subbase with geosynthetics is aimed at the unification of the subbase parameters under the flooring and, at the same time, at a uniform distribution of the stresses transferred by the flooring onto the subbase, which results in a uniform settlement. The solution of the thickness is always related to the specific properties of stabilising geogrids.

#### 3.5.7. Example of Paved Road Subbase Stabilisation

White et al. [150] studied the performance of a geogrid-stabilised aggregate base by analysing the results of cyclic and static automated plate load tests (APLTs) (Figure 29). In their tests, they used APLTs to determine the in situ resilient modulus (*M_r_*) based on a load of 1000 cycles and to study the changes in the *M_r_* and permanent deformation with loading cycles based on 10,000 cycles. The results show that the in situ *M_r_* values for the geogrid-stabilised aggregate subbase layer are, on average, about 10 times higher than the subgrade layer. The results demonstrated an interesting approach to the mechanistic characterisation of the pavement foundation systems, which is of interest to engineers using mechanistic-empirical pavement design methods to a design-optimised pavement with stabilised subbase layers.

## 4. Conclusions

Geosynthetics are polymeric products which are applied to fulfil various functions in civil and geotechnical engineering. Since geosynthetics can perform different functions, as presented in the first part of this paper, they should be designed to meet criteria to sufficiently perform these functions in given applications. The filtration and stabilisation functions were also discussed in this paper.

As presented in parts two and three of this study, geosynthetics for both filtration and stabilisation functions require specific features, since their application is quite complex. Whilst filtration seems to be more established with standardised design approaches and many successful applications, stabilisation is still at a relatively initial point on the learning curve, especially in the context of differentiation from well-established reinforcing applications. The authors hope that the review given in this paper will help to put in order the variability in the specific applications of geosynthetics for both of the functions.

## Figures and Tables

**Figure 1 polymers-14-05492-f001:**
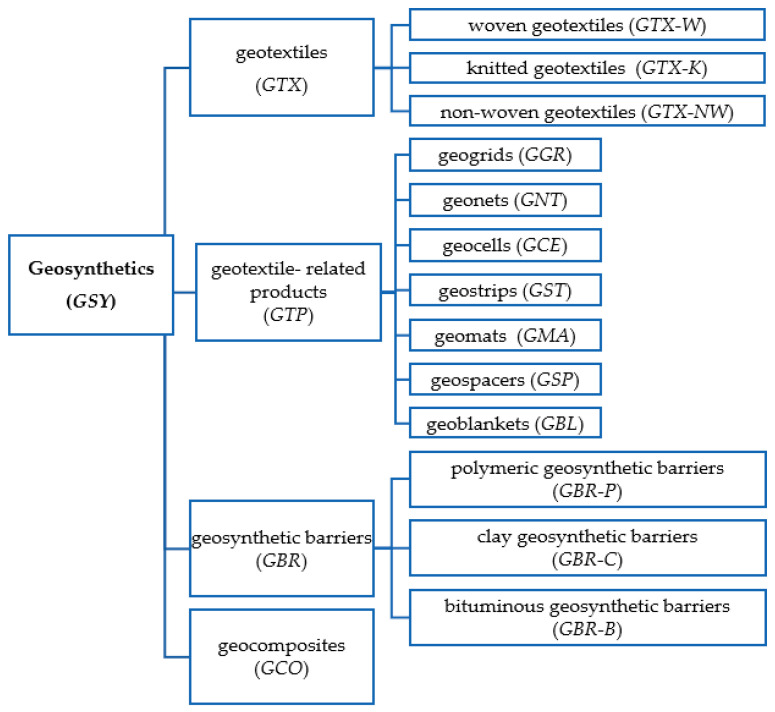
Types of geosynthetics (adapted from [1]).

**Figure 2 polymers-14-05492-f002:**
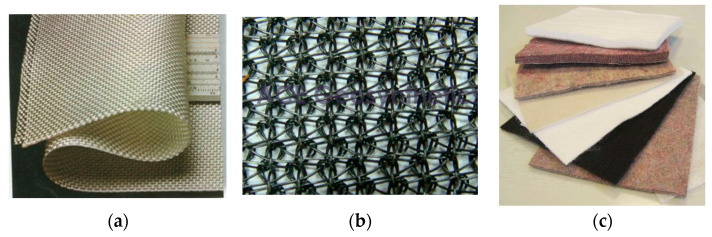
Examples of: (**a**) woven geotextile; (**b**) knitted geotextile; (**c**) nonwoven geotextile.

**Figure 3 polymers-14-05492-f003:**
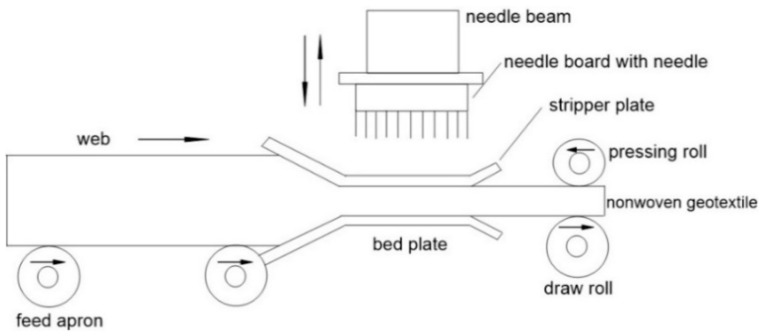
Scheme of needle-punching process (adapted from [12]).

**Figure 4 polymers-14-05492-f004:**
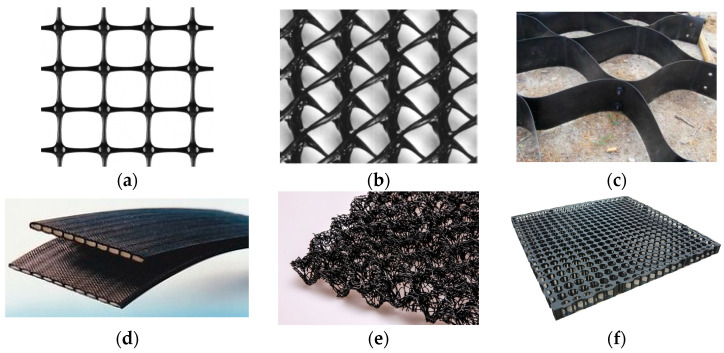
Examples of: (**a**) geogrid; (**b**) geonet; (**c**) geocell; (**d**) geostrip; € geomat; (**f**) geospacer.

**Figure 5 polymers-14-05492-f005:**
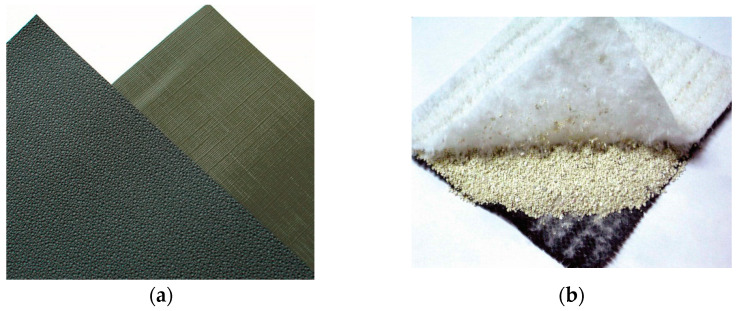
Examples of: (**a**) geomembrane; (**b**) geosynthetic clay layers.

**Figure 6 polymers-14-05492-f006:**
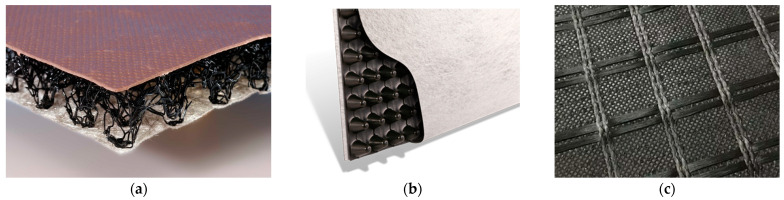
Examples of geocomposites: (**a**) drainage geocomposite; (**b**) drainage geocomposite; (**c**) reinforcement geocomposite.

**Figure 7 polymers-14-05492-f007:**
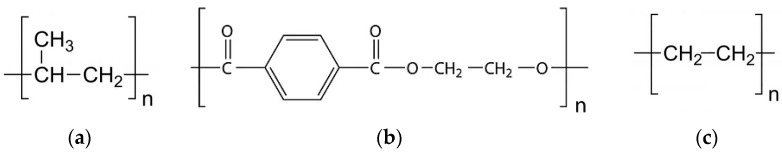
Polymer molecular structure of (**a**) PP; (**b**) PET; (**c**) PE.

**Figure 8 polymers-14-05492-f008:**
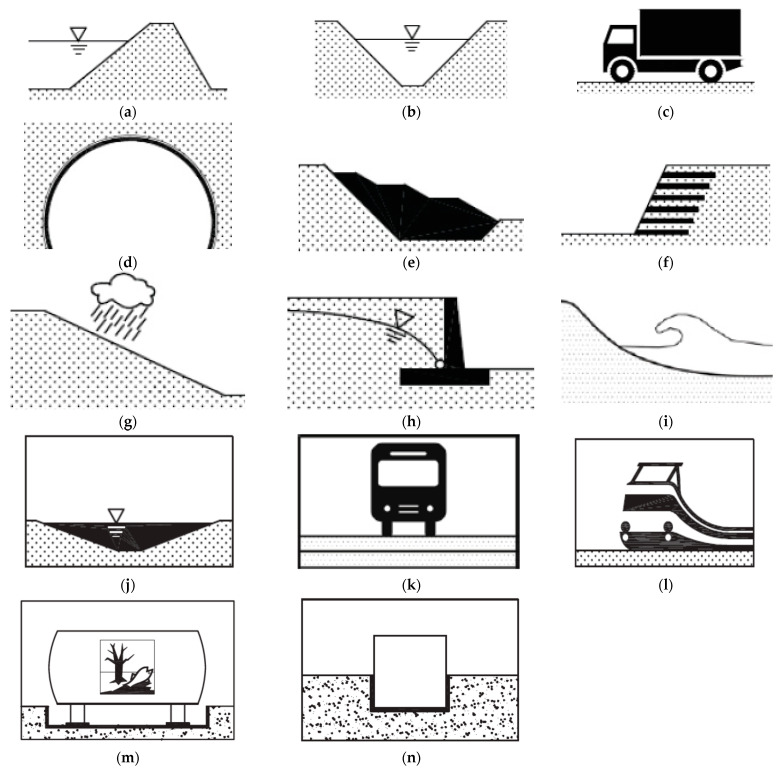
Applications of geosynthetics: (**a**) reservoirs and dams; (**b**) canal; (**c**) road; (**d**) tunnel and underground structures; I landfill; (**f**) retaining wall; (**g**) surface erosion control system; (**h**) drainage system; (**i**) coastal protection; (**j**) liquid waste; (**k**) asphalt reinforcement; (**l**) railways; (**m**) secondary containment; (**n**) waterproofing and underground structures [47,48].

**Figure 9 polymers-14-05492-f009:**
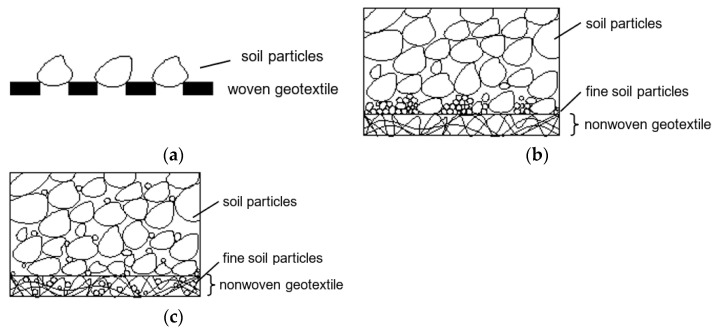
Blocking (**a**); blinding (**b**); and internal clogging (**c**) of geotextile filter.

**Figure 10 polymers-14-05492-f010:**
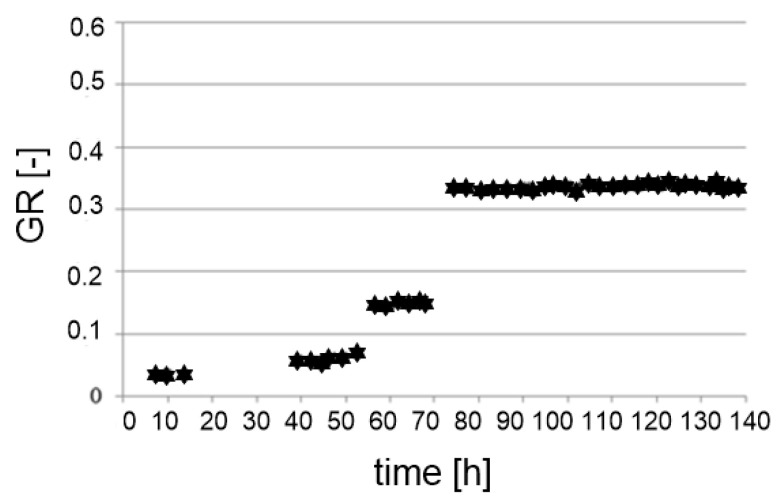
Gradient ratio versus time (adapted from [54]).

**Figure 11 polymers-14-05492-f011:**
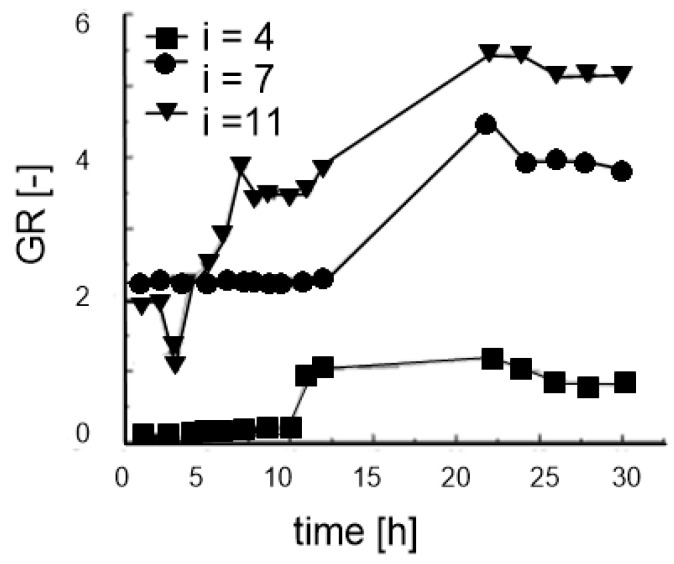
Gradient ratio versus time and hydraulic gradient (adapted from [78]).

**Figure 12 polymers-14-05492-f012:**
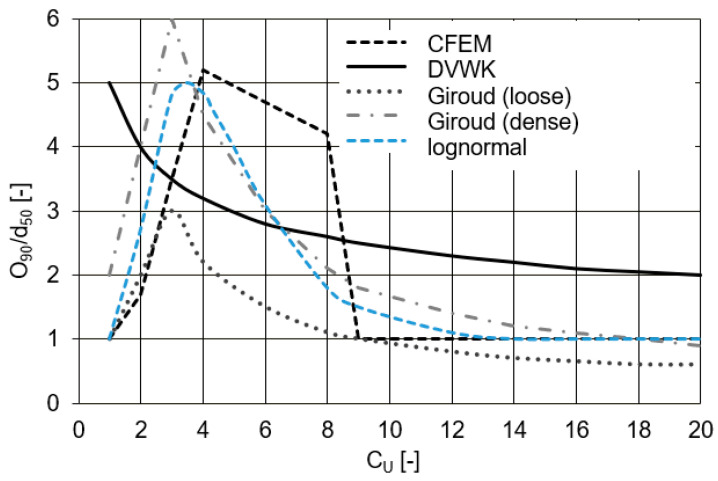
Comparison of retention criteria (adapted from [104]).

**Figure 13 polymers-14-05492-f013:**
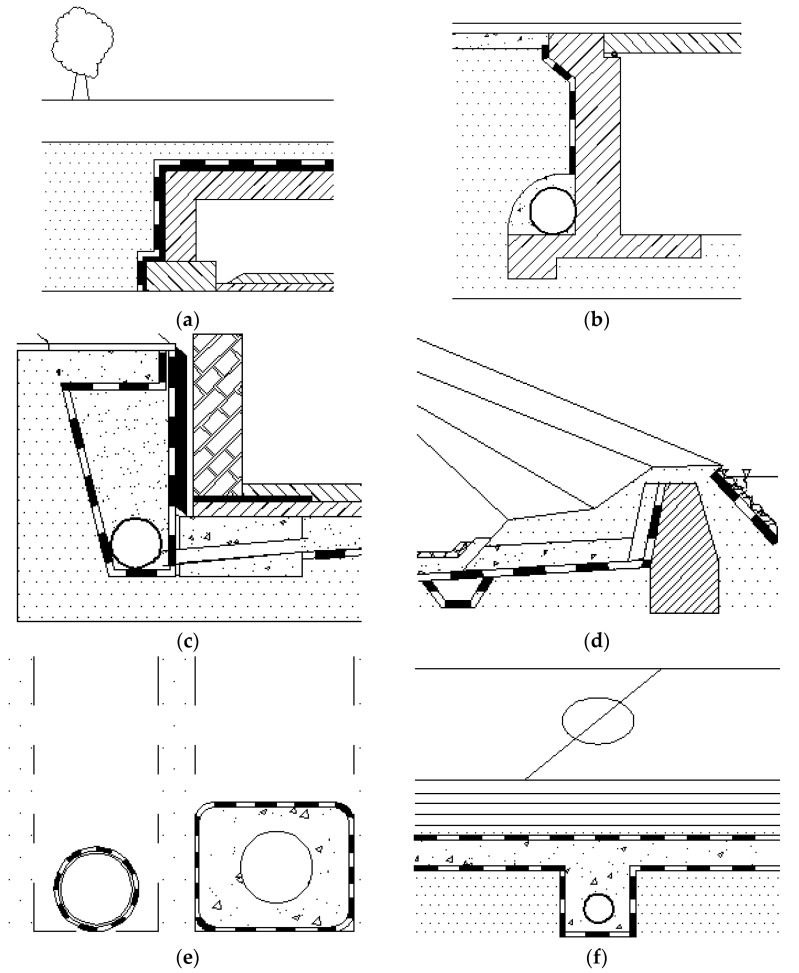
Selected applications of geotextile filters: (**a**) tunnels; (**b**) bridgeheads; (**c**) foundation drainage systems; (**d**) dams; € underdrains; (**f**) sports drainage systems [48].

**Figure 14 polymers-14-05492-f014:**
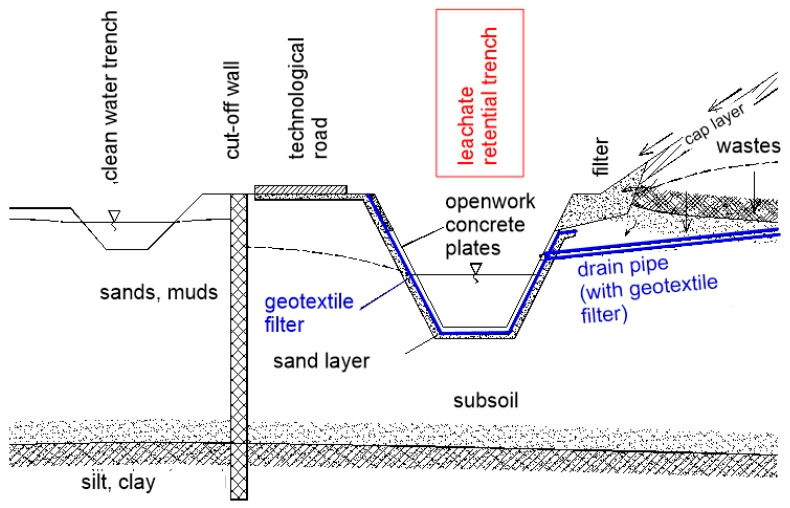
Radiowo landfill: drainage ditches construction.

**Figure 15 polymers-14-05492-f015:**
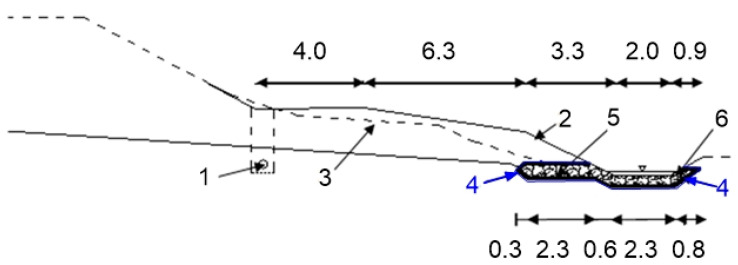
Białobrzegi earthfill dam: drainage system construction (adapted from [48]); 1—old drainage; 2—ground level after renovation; 3—ground level before renovation; 4—nonwoven geotextile; 5—stone drainage (thickness 0.3 m); 6—crush stone (thickness 0.2 m).

**Figure 16 polymers-14-05492-f016:**
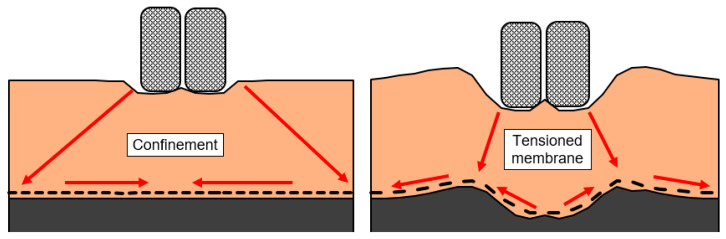
Two mechanisms providing support stabilisation (via confinement) (**left**) and reinforcement (via tensioned membrane (**right**)) (adapted from [123]).

**Figure 17 polymers-14-05492-f017:**
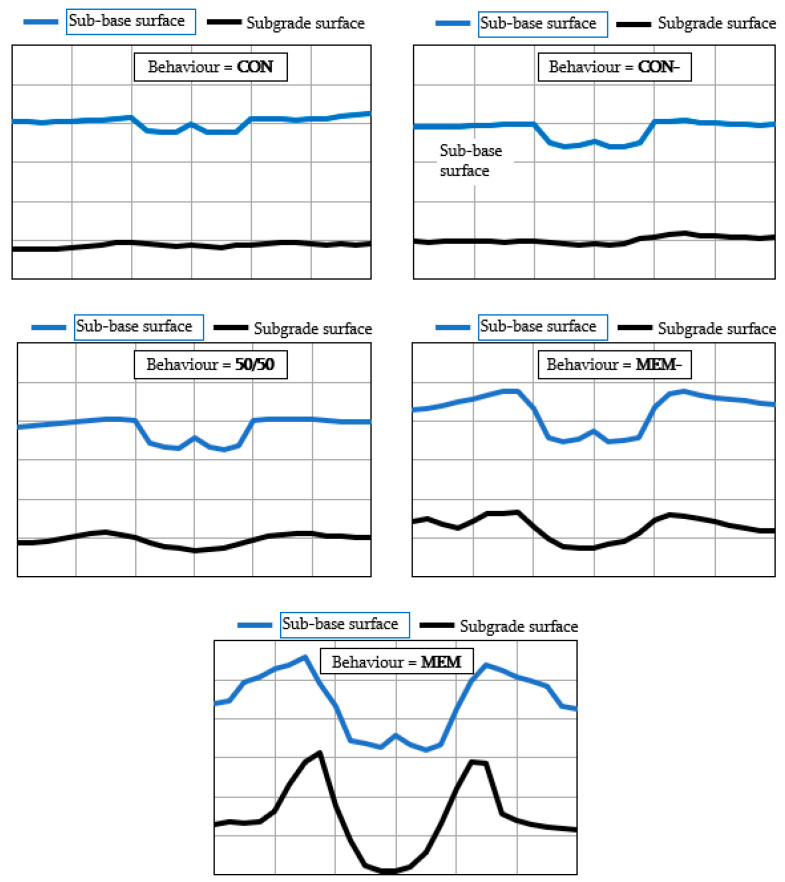
Different types of deformations on sub-base and subgrade surfaces (adapted from [123]).

**Figure 18 polymers-14-05492-f018:**
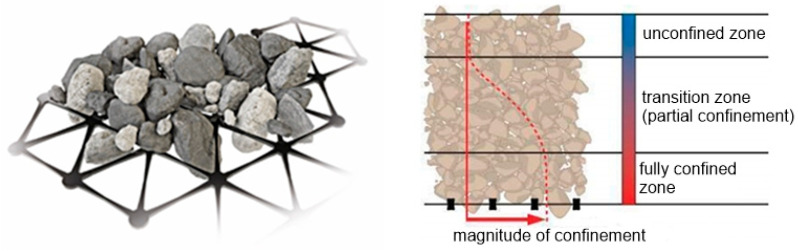
Confinement of particles provided by aggregate interlocking in geogrid apertures (**left**) and its magnitude distribution [124].

**Figure 19 polymers-14-05492-f019:**
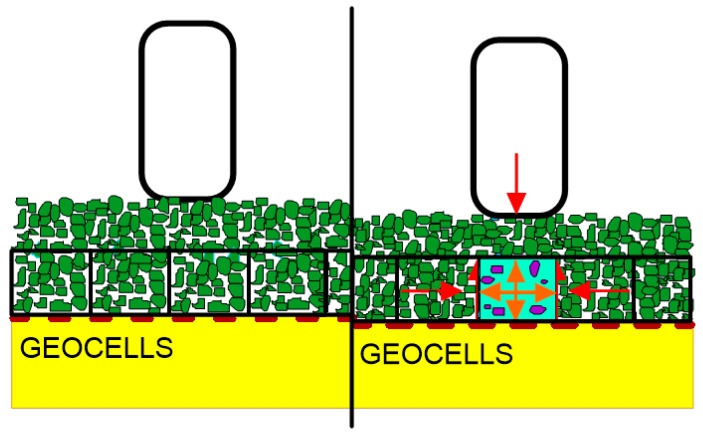
Stabilising mechanism in geocells (adapted from [125]).

**Figure 20 polymers-14-05492-f020:**
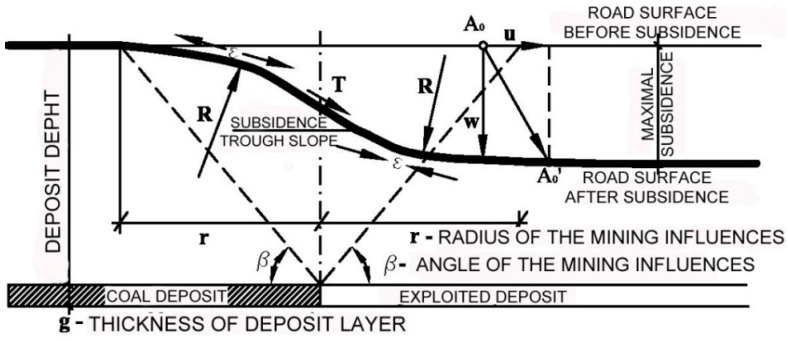
Influence from mining activity on road pavement [44].

**Figure 21 polymers-14-05492-f021:**
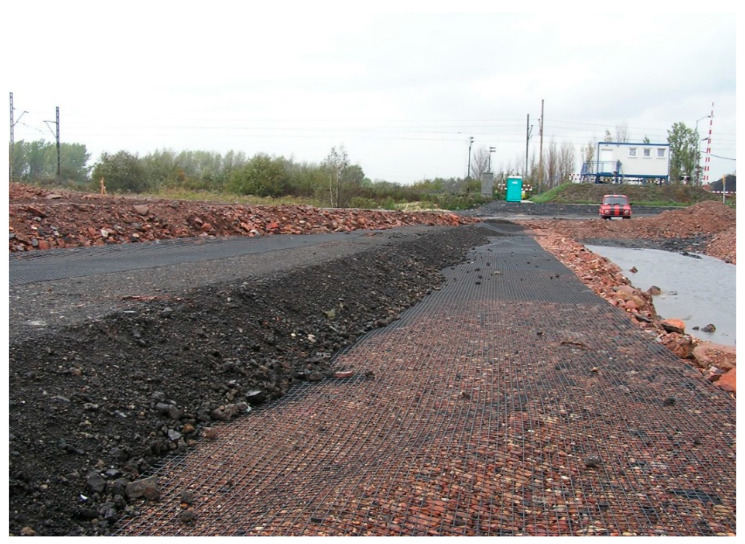
Example of geomattress formed from geogrids and recycled aggregate (mining wastes).

**Figure 22 polymers-14-05492-f022:**
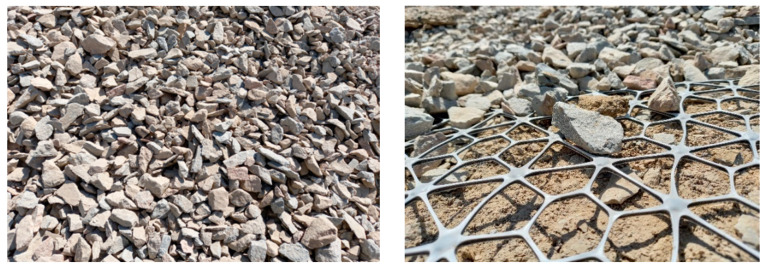
Aggregate (**left**) and geogrid (**right**) used to construct stabilised working platform [139].

**Figure 23 polymers-14-05492-f023:**
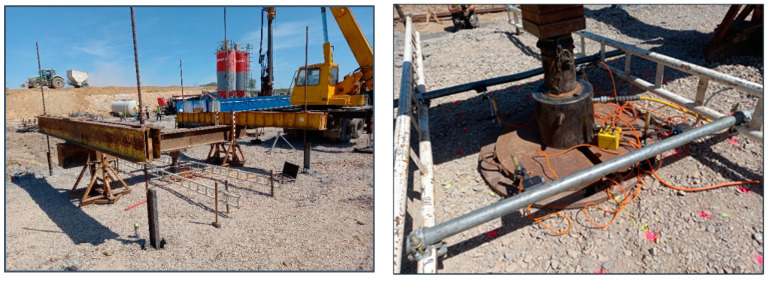
Plate load test to determine bearing capacity of stabilised working platform [139].

**Figure 24 polymers-14-05492-f024:**
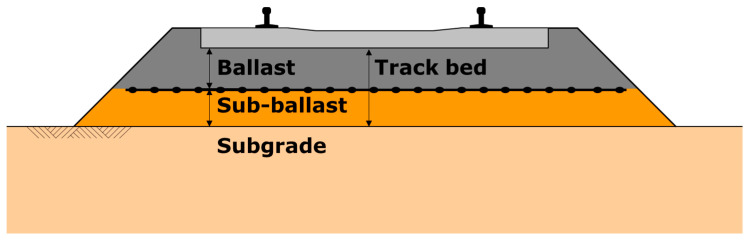
Location of the geogrid for ballast stabilisation [141].

**Figure 25 polymers-14-05492-f025:**
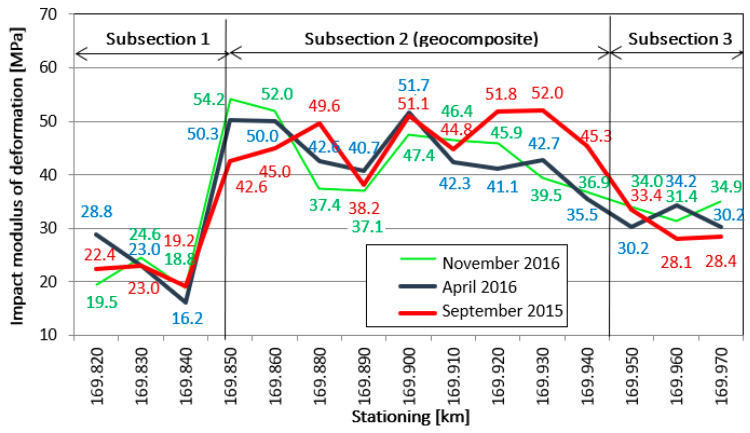
Measurement of modulus along observed track on different sections (adapted from [144]).

**Figure 26 polymers-14-05492-f026:**
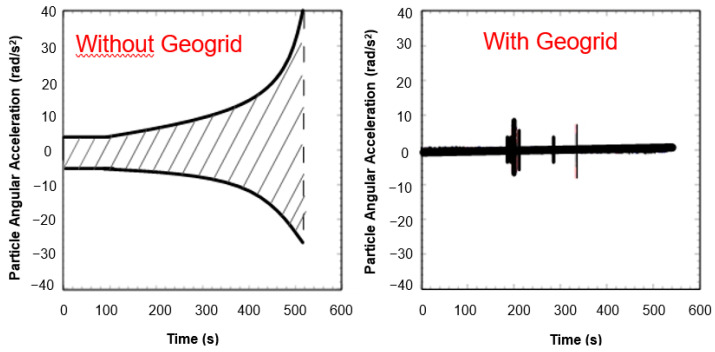
Reduction in particle angular acceleration over time (adapted from [145]).

**Figure 27 polymers-14-05492-f027:**
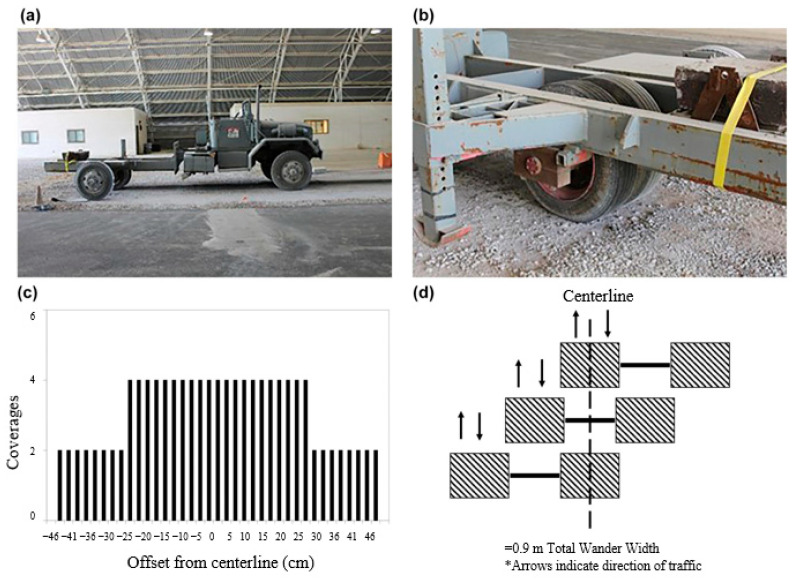
Traffic application equipment and traffic pattern: (**a**) overall view of load cart; (**b**) close-up of test gear configuration; (**c**) normally distributed pattern; (**d**) traffic application pattern [126].

**Figure 28 polymers-14-05492-f028:**
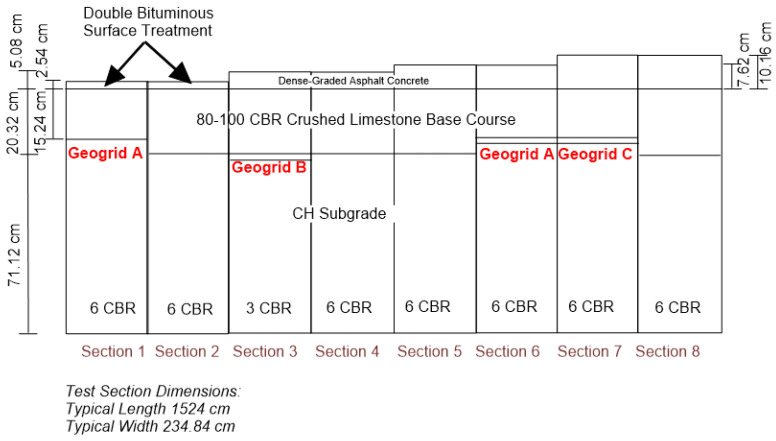
ERDC test sections build-up (adapted from [148]).

**Figure 29 polymers-14-05492-f029:**
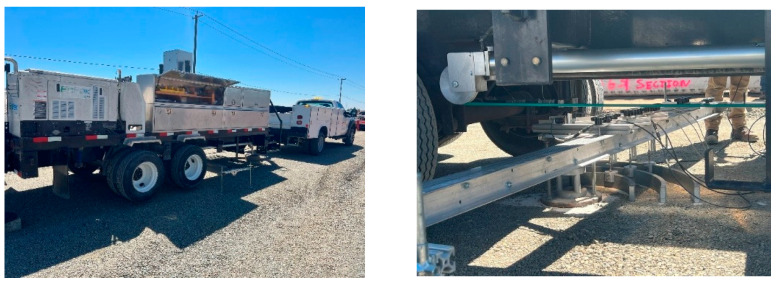
Automated plate load tests (APLT) used to determine in situ resilient modulus of aggregate layer.

**Table 1 polymers-14-05492-t001:** Polymers used for geosynthetics manufacturing.

Geosynthetics	Polymers
Geotextiles	PP, PET, PE, PA
Geotextile-related products	PP, PET, HDPE, MDPE
Geosynthetic barriers	PP, PVC, HDPE, LLDPE, VLDPE, CSPE
Geocomposites	PE, PP, PVC, PS

**Table 2 polymers-14-05492-t002:** Resistance of commonly used polymers in geosynthetics.

Factors	Resistance of Polymers		
	PP	PET	PE
UV stabilised	+++ ^1^	+++	+++
Acids	+++	+	+++
Alkalis	+++	+	+++
Salts	+++	+++	+++
Detergents	+++	+++	+++
Hydrolysis	+++	+++	+++
Steam	+	+	+
Heat	++	+++	+
Creep	+	+++	+
Microorganisms	+++	+++	+++

^1^ +++ high; ++ medium; + low.

**Table 3 polymers-14-05492-t003:** Geosynthetics and their primary functions.

Functions	Geosynthetics
Filtration	Geotextiles, geocomposites
Drainage	Geotextiles, geonets, geocomposites
Reinforcement Separation	Geotextiles, geogrids, geocomposites Geotextiles, geocomposites
Barrier	Geomembranes, geosynthetic clay liners, geocomposites
Protection	Geotextiles, geocomposites
Containment Stabilisation	Geobags, geotubes Geogrids, geocells

**Table 4 polymers-14-05492-t004:** Selected components of artificial leachate used in laboratory tests.

Authors (Year)	Leachate Composition	Type of Tests
Rowe (2005) [69]	Mainly CaCO_3_; 29–34% Ca	Laboratory
Rowe and McIsaac (2005) [70]	>50% CaCO_3_, 16–21% Si, <8% Fe, 5% Mn	Field
Yu and Rowe (2012) [71]	Inorganic solid compounds	Bio-clog 2D model
Wu et al. (2018) [72]	>55% Ca, 9% Si, 9% Fe, 7% Zn	Laboratory

**Table 5 polymers-14-05492-t005:** Selected retention criteria.

Authors (Year)	Criteria	Comments
Zitcher (1975) [93]	O_50_ = (2.5 ÷ 3.7)d_50_	-
Sweetland (1977) [94]	O_15_ ≤ d_85_	-
Schober and Teindl (1979) [95]	2.5 ≤ O_90_/d_50_ ≤ 4.5	C_U_ ^1^ ≤ 5; t_GTX_ ^2^ < 1 mm
	4.5 ≤ O_90_/d_50_ ≤ 7.5	C_U_ ≤ 5; t_GTX_ > 2 mm
	1.0 ≤ O_90_/d_50_ ≤ 2.8	5 < C_U_ ≤ 20
Rankilor (1981) [96]	< d_85_	0.02 ≤ d_85_ ≤ 0.25 mm
	> d_15_	d_85_ > 0.25 mm
Giroud (1982) [97]	$1\;<\;C_{U}^{\prime }$ ^3^ < 3	
	$O_{95}\;<\;C_{U}^{\prime }$d_50_	I_D_ ^4^ ≤ 35%
	$O_{95}\;<\;1.5C_{U}^{\prime }$d_50_	35% < I_D_ < 65%
	$O_{95}\;<\;2C_{U}^{\prime }$d_50_	I_D_ ≥ 65%
	$C_{U}^{\prime }$ ≥ 3	
	$O_{95}\;<\;(9/C_{U}^{\prime })$d_50_	I_D_ ≤ 35%
	$O_{95}\;<\;(13.5/C_{U}^{\prime }$)d_50_	35% < I_D_ < 65%
	$<\;(18/C_{U}^{\prime }$)d_50_	I_D_ ≥ 65%
Carroll (1983) [98]	O_95_ ≤ (2 ÷ 3)d_85_	-
Christopher and Holtz (1985) [99] FWHA (1998) [100]	C_U_ < 2 or C_U_ > 8	sand laminar water flow
	O_95_ ≤ d_85_	
	2 ≤ C_U_ ≤ 4	
	O_95_ ≤ 0,5 C_U_ d_85_	
	4 < C_U_ ≤ 8	
	O_95_ ≤ (8/C_U_) d_85_	
	O_95_ < d_15_	sand turbulent water flow
	O_50_ < 0.5 d_85_	
	O_95_ < 1.8 d_85_	silt, clay turbulent water flow
	O_95_ ≤ 3 mm	
Rollin et al. (1990) [101]	O_95_ ≤ (1 ÷ 1.5)d_85_	-
Corbet (1993) [102]	O_90_ = (1 ÷ 3)d_90_	1 ≤ C_U_ ≤ 5
	O_90_ < (1 ÷ 3)d_90_	5 < C_U_ < 10t_GTX_ ≤ 2 mm
	O_90_ = (1.8 ÷ 6)d_50_	5 < C_U_ < 10t_GTX_ > 2 mm
Lafleur (1999) [103]	O_95_ < (1 ÷ 5)d_30_	internal unstable soils
Giroud (2010) [85]	C_U_ ≤ 3	
	O_95_ ≤ ($C_{U}^{\prime }$)^0.3^$d_{85S}^{\prime }$ ^5^	I_D_ ≤ 35%
	O_95_ ≤ 1.5($C_{U}^{\prime }$)^0.3^$d_{85S}^{\prime }$	35% < I_D_ < 65%
	O_95_ ≤ 2($C_{U}^{\prime }$)^0.3^$d_{85S}^{\prime }$	I_D_ ≥ 65%
	C_U_ ≥ 3	
	O_95_ ≤ 9$d_{85S}^{\prime }$/$(C_{U}^{\prime }$)^1.7^	I_D_ ≤ 35%
	O_95_ ≤ 13.5$d_{85S}^{\prime }$/$(C_{U}^{\prime }$)^1.7^	35% < I_D_ < 65%
	O_95_ ≤ 18$d_{85S}^{\prime }$/$(C_{U}^{\prime }$)^1.7^	I_D_ ≥ 65%

^1^ C_U_—coefficient of uniformity; ^2^ t_GTX_—geotextile thickness; ^3^
$C_{U}^{\prime }$—coefficient of uniformity in a linear particle size distribution; ^4^ I_D_—relative density; ^5^
$d_{85S}^{\prime }—$soil particle size (d_85_) according to the linear particle size distribution.

**Table 6 polymers-14-05492-t006:** Typical values of the hydraulic gradient in soil next to the geotextile filter (adapted from [85]).

Applications	Hydraulic Gradient
Liquid reservoir with GCL	>10
Dam toe drain	2.0
Landfill drainage layer	1.5
Road edge drain	≤1.0
Dewatering trench	≤1.0

**Table 7 polymers-14-05492-t007:** Propositions for calculating the number of constrictions (adapted from [115]).

Author (Year)	Proposition
Elsharief and Lovell (1996) ^1^	$m=\frac{t_{\mathrm{GTX}}}{O_{98}}$
Urashima and Vidal (1998) ^2^	$m=\frac{t_{\mathrm{GTX}}}{s}$
Giroud (1996, 2010) ^3^	m = $\frac{t_{\mathrm{GTX}}}{d_{c}}$

^1^ O_98_—geotextile opening diameter for which 98% of the remaining openings are smaller than that value; ^2^ s—constriction distance evaluated through a retro-analysis process; ^3^ d_c_—constriction distance: d_c_ = d_f_/$\sqrt{1-n}$ (d_f_—fibre diameter).

**Table 8 polymers-14-05492-t008:** Coefficients water permeability for nonwoven geotextiles exploited at the Radiowo landfill leachate drainage system (adapted from [68]).

	Permeability Coefficient (m/s)			
Nonwoven Geotextile	Load (kPa)			
	0	2	20	200
Unworn	0.0080	0.0034	0.0027	0.0014
Worn after 12 years of exploitation	0.0029	0.0019	0.0016	0.0008
Worn after 15 years of exploitation	0.0020	0.0011	0.0007	0.0004

**Table 9 polymers-14-05492-t009:** Coefficients water permeability for nonwoven geotextiles exploited at the Białobrzegi earthfill dam (adapted from [118]).

	Permeability Coefficient (m/s)			
Nonwoven Geotextile	Load (kPa)			
	0	2	20	200
Unworn	0.00268	n/a ^1^	n/a	n/a
After 23 years of exploitation	0.00104	0.00073	0.00055	0.00032

^1^ n/a—not available.

**Table 10 polymers-14-05492-t010:** List of most popular applications where stabilising geosynthetics provide improvement to performance of aggregate layers.

Application	Specific Aggregate Layer Stabilised
Road—unpaved	Stabilisation of base
Road—unpaved	Stabilisation of subbase
Road—paved	Stabilisation of base
Road—paved	Stabilisation of subbase
Railway	Stabilisation of ballast
Railway	Stabilisation of sub-ballast
Runway	Stabilisation of subbase
Working platform	Stabilisation of platform
Liquefaction protection	Gravel raft stabilisation
Industrial floor	Bearing stratum (base) stabilisation
Mining subsidence protection	Stabilisation of geomattresses
Foundations	Stabilisation of regulating layer
Surcharge reduction on organic formations	Lightweight fill stabilisation
Rayleigh waves mitigation	Stabilisation of fill replacement

## Data Availability

Not applicable.

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
