# Peer review of "Geosynthetics for Filtration and Stabilisation: A Review"

_polymers, 2022, doi:10.3390/polym14245492_

Round 1
Reviewer 1 Report
This work deals with the use of geosynthetics in civil and environmental engineering. These polymer-based products have had a wide diffusion in the last decades due to the economic convenience in the prefabrication of artifacts able to absolve many function in the field of civil, industrial and environmental works. After an initial state of the art on the types of geosynthetics and their field of use, the knowledge in the fields of filtration and stabilization is summarized. The paper is completed by two case histories of geosynthetic application as drainage in Poland and by some cases of base reinforcement of paved and unpaved roads, railway and industrial floor.
This paper may be considered a state of the art of geosynthetics uses in filtration and reinforcement purposes and can be published as technical note, but a further revision is necessary after the reduction in length of the text. For example, figs 8, 9, 10, 15, 16, 18, 33 and 34 can be removed being useless.
Suggestions:
Add the acronym PE in Polyethylene at row 107;
Add containment function (geo-tubes, geo-bags) in table 3;
Add the term sludge at row 172;
Do not use the term piping at row 207 because it is inappropriate, use the term internal soil stability;
Explain the effect in time of air bubble accumulation during a filtration process;
Explain what means O98, O95, O90, O50, O15, d90, d85, d50, d30, d15.
Author Response
Dear Reviewer,
We would like to express our appreciations for all your effort put in reviewing the manuscript “Geosynthetics for Filtration and Stabilisation: A Review”. Thank you for your detailed comments and suggestions. We would like to assure you that the content of the paper has been carefully reviewed and improved according to all the comments and suggested amendments. We confirm that English language were checked by a professional translator. We hope we have responded satisfactory to your concerns. Below you can find specific answers to all recommended improvements.
Comment: This paper may be considered a state of the art of geosynthetics uses in filtration and reinforcement purposes and can be published as technical note, but a further revision is necessary after the reduction in length of the text. For example, figs 8, 9, 10, 15, 16, 18, 33 and 34 can be removed being useless.
Answer: Authors agree that the text could be shortened. Figures 9, 10, 16, 18, 33 were removed. In our opinion, figures 8, 15 and 34 are valuable.
Comment: Add the acronym PE in Polyethylene at row 107.
Answer: The acronym PE was added.
Comment: Add containment function (geo-tubes, geo-bags) in table 3.
Answer: It was added.
Comment: Add the term sludge at row 172.
Answer: The term sludge was added.
Comment: Do not use the term piping at row 207 because it is inappropriate, use the term internal soil stability.
Answer: Authors do not agree with this statement. Filter design is a balancing act between piping and clogging. According to Fannin J. (2007), in Geosynthetics in Civil Engineering: “Filtration compatibility is predicated on the geotextile satisfying a requirement for soil retention. Incompatibility may take the form of unacceptable piping or clogging. Piping refers to particle migration through the geotextile, whereas clogging is a result of entrapment of particles within the geotextile”. Piping mechanism was studied also by Giroud (2010), Development of criteria for geotextile and granular filters, Proceedings of the 9th International Conference on Geosynthetics, Guarujá, Brazil. Recently, the clogging, blinding and piping levels of the filtering systems have been evaluated by Moraci et al. (2022): Soil/geotextile filter compatibility: a geometrical, experimental and micro-structural approach, Geosynthetics International, ahead of print.
Comment: Explain the effect in time of air bubble accumulation during a filtration process.
Answer: Air bubbles can be entered into the voids and block the water flow with time. For that reason air bubbles formation should be prevented. In the gradient ratio test, the system is rested overnight to saturate soil-geotextile system very well. Before starting the test, it is important to check the manometer tubes whether there is any air bubbles or not. The flow saturation procedure is also preferred to prevent soil migration through the tested geotextile. Moreover, air clogging can be decreased by using deaired water. It is essential that use of deaired water in the experiment reduce the risk of forming air bubbles within the test set-up.
Comment: Explain what means O98, O95, O90, O50, O15, d90, d85, d50, d30, d15.
Answer: O98 (or FOS – Filtration Opening Size)- the pore diameter for which 98% of the remaining pore diameters are smaller; O95 (or AOS – Apparent Opening Size)- the pore diameter for which 95% of the remaining pore diameters are smaller; O90 (or Characteristic Opening Size)- the pore diameter for which 90% of the remaining pore diameters are smaller; O50- the pore diameter for which 50% of the remaining pore diameters are smaller; O15- the pore diameter for which 15% of the remaining pore diameters are smaller; d90- the grain size diameter of which 90% of the distribution is finer; d85- the grain size diameter of which 85% of the distribution is finer; d50- the grain size diameter of which 50% of the distribution is finer; d30- the grain size diameter of which 30% of the distribution is finer; d15- the grain size diameter of which 15% of the distribution is finer.
All symbols were explained in the text.
We would like to inform that 2 references were removed:
- Koda, E.; Paprocki, P. Durability of leachate drainage systems of old sanitary landfills. Proceedings of the 3rd International Conference “Geofilters”, Warsaw, Poland, 5-7 June 2000, pp. 215–222.
- Krzywosz, Z.; Koda, E. Badania parametrów technicznych wÅ‚óknin przewidzianych do wbudowania jako warstwy filtracyjne. Nadzór autorski remontu zapory bocznej “BiaÅ‚obrzegi”, GEOTEKO Sp. z o.o., 1994. [In Polish]
The following references have been added:
- Koda, E.; Szymanski, A.; Wolski, W. Field and laboratory experience with the use of strip drains in organic soils. Can. Geotech. J. 1993, 30(2), 308–318.
- Koda, E.; Kiersnowska, A.; Kawalec, J.; OsiÅ„ski, P. Landfill Slope Stability Improvement Incorporating Reinforcements in Reclamation Process Applying Observational Method. Appl. Sci. 2020, 10(5), 1–14.
- Parthiban, D.; Vijayan, D.S.; Koda, E.; Vaverková, M.D.; Piechowicz, K.; OsiÅ„ski, P.; Duc, B.V. Role of industrial based precursors in the stabilization of weak soils with geopolymer – A review. Case Stud. Constr. Mater. 2022, 16, 1–17, doi.org/ 10.1016/j.cscm.2022.e00886.
With our best regards,
Authors

Reviewer 2 Report
This review is written in accordance with the requirements and propositions of the Polymers scientific journal. It covers a wide range of geomaterials that can be made from already-used and recycled materials. The images are well-composed and have a clear representation of the main text. The references used are satisfactory.
It is an interesting approach and it is good that the resistance of the polymer was used because it is an important property to use these materials.
The naming proposal to call such materials geosynthetic is interesting and this name includes the given form of this review.
Pictures or graphs 12 and 13 should be of better resolution and clearer.
The text presents the use and application properties of these materials very well.
Authors should correct minor spelling errors in the text.
Author Response
Dear Reviewer,
We would like to express our appreciations for all your effort put in reviewing the manuscript “Geosynthetics for Filtration and Stabilisation: A Review”. Thank you for your detailed comments and suggestions. We would like to assure you that the content of the paper has been carefully reviewed and improved according to all the comments and suggested amendments. We confirm that English language were checked by a professional translator. We hope we have responded satisfactory to your concerns. Below you can find specific answers to all recommended improvements.
Comment: Pictures or graphs 12 and 13 should be of better resolution and clearer.
Answer: Pictures 12 and 13 were improved.
Comment: Authors should correct minor spelling errors in the text.
Answer: Authors confirm that English language were checked by a professional translator. The minor spelling errors were corrected.
We would like to inform that 2 references were removed:
Koda, E.; Paprocki, P. Durability of leachate drainage systems of old sanitary landfills. Proceedings of the 3rd International Conference “Geofilters”, Warsaw, Poland, 5-7 June 2000, pp. 215–222.
Krzywosz, Z.; Koda, E. Badania parametrów technicznych wÅ‚óknin przewidzianych do wbudowania jako warstwy filtracyjne. Nadzór autorski remontu zapory bocznej “BiaÅ‚obrzegi”, GEOTEKO Sp. z o.o., 1994. [In Polish]
The following references have been added:
Koda, E.; Szymanski, A.; Wolski, W. Field and laboratory experience with the use of strip drains in organic soils. Can. Geotech. J. 1993, 30(2), 308–318.
Koda, E.; Kiersnowska, A.; Kawalec, J.; OsiÅ„ski, P. Landfill Slope Stability Improvement Incorporating Reinforcements in Reclamation Process Applying Observational Method. Appl. Sci. 2020, 10(5), 1–14.
Parthiban, D.; Vijayan, D.S.; Koda, E.; Vaverková, M.D.; Piechowicz, K.; OsiÅ„ski, P.; Duc, B.V. Role of industrial based precursors in the stabilization of weak soils with geopolymer – A review. Case Stud. Constr. Mater. 2022, 16, 1–17, doi.org/ 10.1016/j.cscm.2022.e00886.
With our best regards,
Authors

Reviewer 3 Report
You initially described geosynthetics in detail and then described geosynthetics in terms of both filtration and stabilization. The structure of your review is relatively complete. Your review is rich in references, pictures and tables that will give the reader a clearer picture of the field. I agree to accept your manuscript.
Author Response
Dear Reviewer,
We would like to express our appreciations for all your effort put in reviewing the manuscript “Geosynthetics for Filtration and Stabilisation: A Review”. We highly appreciate the careful review. We want to confirm that English language were checked by a professional translator. Thank you very much for your kind words about our manuscript.
We would like to inform that 2 references were removed:
Koda, E.; Paprocki, P. Durability of leachate drainage systems of old sanitary landfills. Proceedings of the 3rd International Conference “Geofilters”, Warsaw, Poland, 5-7 June 2000, pp. 215–222.
Krzywosz, Z.; Koda, E. Badania parametrów technicznych wÅ‚óknin przewidzianych do wbudowania jako warstwy filtracyjne. Nadzór autorski remontu zapory bocznej “BiaÅ‚obrzegi”, GEOTEKO Sp. z o.o., 1994. [In Polish]
The following references have been added:
Koda, E.; Szymanski, A.; Wolski, W. Field and laboratory experience with the use of strip drains in organic soils. Can. Geotech. J. 1993, 30(2), 308–318.
Koda, E.; Kiersnowska, A.; Kawalec, J.; OsiÅ„ski, P. Landfill Slope Stability Improvement Incorporating Reinforcements in Reclamation Process Applying Observational Method. Appl. Sci. 2020, 10(5), 1–14.
Parthiban, D.; Vijayan, D.S.; Koda, E.; Vaverková, M.D.; Piechowicz, K.; OsiÅ„ski, P.; Duc, B.V. Role of industrial based precursors in the stabilization of weak soils with geopolymer – A review. Case Stud. Constr. Mater. 2022, 16, 1–17, doi.org/ 10.1016/j.cscm.2022.e00886.
With our best regards,
Authors

Round 2
Reviewer 1 Report
The authors have made some, but not all, of the modifications indicated by me, and they do not accompany the text with a note replying to my observations.
Therefore, the following changes are necessary and another revision by me is also necessary:
Fig. 3, Fig. 7, Fig. 8, Fig. 13, Fig. 21, Fig 23 and Fig. 29: the figures must be eliminated because they add no new knowledge to the text.
Row 218 – Don’t use the term piping because it is inappropriate, use the term internal soil stability;
Explain the effect in time of air bubble accumulation during an experimental filtration process;
Fig 6 specify in the text and also in the caption the typology of these geocomposites;
Figg. 16 and 19 must be reduced and simplified.
Moreover, all the figures should be edited according to the standard of the journal as in the present form they do not seem to be well readable.
Author Response
Dear Reviewer,
We would like to express our appreciations for all your effort put in reviewing the manuscript “Geosynthetics for Filtration and Stabilisation: A Review”. Thank you for your detailed comments and suggestions. We would like to assure you that the content of the paper has been carefully reviewed. We confirm that English language was checked by a professional translator. Below you can find specific answers to all recommended improvements.
Comment: Fig. 3, Fig. 7, Fig. 8, Fig. 13, Fig. 21, Fig 23 and Fig. 29: the figures must be eliminated because they add no new knowledge to the text.
Answer: In our opinion, figures 3, 7, 8, 13, 21, 23 and 29 are valuable. Figure 3 presents one of the most important nonwoven geotextiles manufacturing process. This scheme makes our paper more interesting. Figure 7 presents the polymer molecular structure of PP, PET, PE- commonly used polymers in geosynthetics. It can be useful for the reader. Figure 8 is based on the acceptable standard. This figure shows the applications of geosynthetics, what should be highlighted in this paper. It should not be also omitted to point out, that Figure 13 is closely related to the topic of the paper. Selected applications of geotextile filter will be very helpful for the reader. Figures: 21, 23 and 29 present interesting example of applications of geosynthetics and in-situ tests. These tests are not commonly known. The reader can view the plate load test to determine bearing capacity of stabilised working platform and automated plate load tests (APLT) used to determine in situ resilient modulus of aggregate layer. In our opinion, in a review paper these figures are needed.
What is important, out of the 8 suggested, 5 figures were removed the last time.
Comment: Row 218 – Don’t use the term piping because it is inappropriate, use the term internal soil stability.
Answer: Our comments were given in the last cover letter. Authors do not agree with this statement. Filter design is a balancing act between piping and clogging. According to Fannin J. (2007), in Geosynthetics in Civil Engineering: “Filtration compatibility is predicated on the geotextile satisfying a requirement for soil retention. Incompatibility may take the form of unacceptable piping or clogging. Piping refers to particle migration through the geotextile, whereas clogging is a result of entrapment of particles within the geotextile”. Piping mechanism was studied also by Giroud (2010): Development of criteria for geotextile and granular filters, Proceedings of the 9th International Conference on Geosynthetics, Guarujá, Brazil. Recently, the clogging, blinding and piping levels of the filtering systems have been evaluated by Moraci et al. (2022): Soil/geotextile filter compatibility: a geometrical, experimental and micro-structural approach, Geosynthetics International, ahead of print.
Comment: Explain the effect in time of air bubble accumulation during an experimental filtration process.
Answer: Our comments were given in the last cover letter. Air bubbles can be entered into the voids and block the water flow with time. For that reason air bubbles formation should be prevented. In the gradient ratio test, the system is rested overnight to saturate soil-geotextile system very well. Before starting the test, it is important to check the manometer tubes whether there is any air bubbles or not. The flow saturation procedure is also preferred to prevent soil migration through the tested geotextile. Moreover, air clogging can be decreased by using deaired water. It is essential that use of deaired water in the experiment reduce the risk of forming air bubbles within the test set-up.
It was added in the text (row 281-284).
Comment: Fig 6 specify in the text and also in the caption the typology of these geocomposites.
Answer: It was added.
Comment: Fig. 16 and 19 must be reduced and simplified.
Answer: Thank you for this comment, however to properly explain the mechanism related to reinforcing function and stabilising mechanism in geocells it is not possible to reduce these schemes more. We are sure that these figures will be understandable and valuable for the reader.
Comment: Moreover, all the figures should be edited according to the standard of the journal as in the present form they do not seem to be well readable.
Answer: All figures were prepared according to journal template.
With our best regards,
Authors
